# Effects of Different Dietary Calcium Levels on the Performance, Egg Quality, and Albumen Transparency of Laying Pigeons

**DOI:** 10.3390/ani9030110

**Published:** 2019-03-22

**Authors:** Lingling Chang, Rui Zhang, Shengyong Fu, Chunyu Mu, Qingping Tang, Zhu Bu

**Affiliations:** Poultry Institute, Chinese Academy of Agricultural Sciences, Yangzhou 225100, China; jqscll@163.com (L.C.); zrjindy@163.com (R.Z.); xnfsy@163.com (S.F.); muchunyu521@126.com (C.M.); tqp0979@163.com (Q.T.)

**Keywords:** laying pigeon, calcium level, performance, egg quality, albumen transparency

## Abstract

**Simple Summary:**

The transparency of pigeon egg albumen is a special phenotypic trait that different from other poultry eggs, and calcium is one of the main factors. This experiment aimed to study the effects of added dietary Ca on production performance, egg quality, and albumen transparency. It will guide producers on how to feed their laying pigeons and provide a scientific basis for special feed research and the development of laying pigeons with transparent eggs.

**Abstract:**

In order to confirm the dietary calcium (Ca) requirement to keep a balance of the production performance and the albumen transparency, the study examined the effects of different dietary Ca levels on the production performance, egg quality, and albumen transparency of laying pigeons. 1080 pigeon pairs were randomly allocated into six treatment groups, each consisting of six replicates of 30 pigeon pairs per treatment. Ca levels of 0.60, 0.90, 1.20, 1.80, 2.40, and 3.00% were fed. The results showed that the laying rate, average egg weight, and feed to egg ratio were significantly influenced by Ca levels (*p* < 0.05). Albumen percentage, albumen height, Haugh unit, and eggshell thickness at 8 w and 16 w were significantly influenced by Ca levels (*p* < 0.05). The *L**, *a**, *b**, and *c** values of cooked albumen at 8 w and 16 w were all significantly influenced by Ca levels (*p* < 0.05). After 16 w of feeding different Ca levels, the percentage of transparent eggs had an early increasing and later decreasing trend. In conclusion, taking the transparency of pigeon eggs as an assessment index and considering production performance secondly, the optimal level of dietary Ca for laying White King pigeons is 0.90%.

## 1. Introduction

Pigeon eggs are praised as “animal ginseng” due to their many ingredients with high nutritional value such as protein, lecithin, vitamins and iron. They are a safe and healthy food source and can be a medicated diet for all ages [1,2]. With the improvement in living standards for humans and the optimization of the consumption structure, pigeon eggs are no longer used just for feasts but are also gradually being used for daily consumption. Pigeon eggs are usually served boiled or in soup. Therefore, the appearance of cooked pigeon eggs has become the main factor affecting human consumption. They are different from other poultry eggs because the transparency of their albumen is a special phenotypic trait of pigeon eggs. This transparent albumen is more popular because of its glittering and translucent appearance, elasticity, and soft taste. To date, many reports exist concerning conventional pigeon egg quality [3], nutritional ingredients [4], and antioxidant capabilities [5]; however, there are few reports concerning the albumen transparency of pigeon eggs. Previous research from our group has shown that the calcium (Ca) levels of transparent albumen are significantly lower than opaque albumen, indicating that it is possible to improve the transparency of pigeon eggs by reducing the dietary Ca levels [6].

The National Research Council has not yet provided any standards for pigeon nutritional requirements [7]. Some research exists concerning energy and protein levels [8], as well as the methionine and lysine requirements of laying pigeons [9]. Not much research regarding pigeon Ca requirements has been reported. Ca is the most prevalent mineral in the body and is important for many physiological processes. It acts as a structural function by supporting material in bones and it has a signaling function, with intracellular Ca functioning as a secondary messenger for certain hormones. Additionally, Ca acts as a coenzyme for clotting factors and can also cause the release of acetylcholine from pre-synaptic terminals of neurons and thereby facilitate the transmission of nerve impulses [10]. A deficiency of Ca can lead to a decrease in the feed conversion rate for laying fowl [11], cause skeletal abnormalities [12], and affect the quality of eggshell [13].

To find a balance that does not affect the production performance and improves the proportion of transparent pigeon eggs, different levels of Ca were added to the diets of pigeons. This experiment aimed to study the effects of added dietary Ca on production performance, egg quality, and albumen transparency. Consequently, it will guide producers on how to feed their laying pigeons and provide a scientific basis for special feed research and the development of laying pigeons with transparent eggs.

## 2. Materials and Methods

This study was conducted from September to December 2017 at Poultry Institute, Chinese Academy of Agricultural Sciences (PI-CAAS), Yangzhou, China. The experimental procedures were approved by the Animal Ethics Committee of the PI-CAAS, and humane animal care and handling procedures were followed throughout the experiment (protocol number: PI-CAAS-2017-09).

### 2.1. Animals and Husbandry

A total of 1080 female-female pairs of healthy American White King pigeons from 1 to 2 years of age with similar body weights (body weight: 524.78 ± 56.43 g) were randomly allocated to one of six treatment groups, each consisting of six replicates of 30 pigeon pairs per replicate. The 36 replicates were randomly distributed in the facility. The study had lasted for 16 weeks after a 1-week adaptive phase. All animals were given free access to water and food. Two female pigeons were housed per cage (50 cm × 60 cm × 45 cm). The cages were equipped with a nest and a perch and were located in a windowed poultry house in a room under a 16 L: 8 D lighting cycle, with a mean daily temperature of 15 ± 5 °C.

### 2.2. Dietary Treatments

The energy, crude protein [8], methionine and lysine levels [9] of the basal diet were referenced by a previous study and are often used by major pigeon farms. Ca levels of 0.60%, 0.90%, 1.20%, 1.80%, 2.40% and 3.00% in a corn-soybean complete pellet feed were fed respectively to six groups of pigeons. Ca/phosphorus was kept at a ratio of 3.71:1 in every group. The composition and nutrient levels of the diets are shown in Table 1.

### 2.3. Productive Performance

Daily feed intake, egg production and egg weight were monitored by electronic scale (Tuoliduo, Suzhou, China) throughout the trial. From these values, the average daily feed intake (g/day), laying rate (%), average egg weight (g) and feed to egg ratio (g/g) were determined. The remaining feed was weighed every week to calculate the average daily feed intake. Body weights of all pigeons were determined after a 12 h fasting period at the beginning and end of the study. Laying rate (%) = laying eggs every day/pigeon number × 100%, feed to egg ratio (g/g) = feed consumption every week/weight of laying eggs every week.

### 2.4. Egg Quality

During the middle (8 weeks, 8 w) and the end (16 weeks, 16 w) of the experiment, 10 pigeon eggs from each treatment were randomly selected to determine egg quality. Eggshell thickness was measured by an Egg Shell Thickness Gauge. Eggshell strength was measured by an Egg Force Reader. The egg shape index was figured out by height/width, with egg height and width being measured by a vernier caliper [14]. Yolk color, albumen height, and the Haugh unit were measured by an Egg Analyzer (ORKA, Jerusalem, Israel). Eggshell, yolk and albumen were separated and weighed, and the percentage of each was calculated based on the weight of the whole egg [15].

### 2.5. Albumen Transparency

During the middle and the end of the experiment, 10 pigeon eggs from each treatment were randomly chosen for the determination of albumen transparency. The eggs were cooked in boiling water for 30 min, and the shell was then stripped after cooling. For each egg, the same portion of the albumen was taken and cut into the same shape (1 cm × 1 cm × 1 cm) once the albumen and yolk were separated. The samples were put onto a hand-held RM200QC Spectrocolourimeter (X-Rite, Bern, Switzerland) and the *L**, *a**, *b**, and *c** were measured. The hue angle (H° = arctangent (*b**/*a**)) was also calculated. Then, according to the range of *L** (≤45, 45 < x < 55, ≥55), the eggs were divided into groups of transparent eggs, translucent eggs and opaque eggs [16].

### 2.6. Statistical Analysis

All data were analyzed by SPSS 20.0 (SPSS Inc., Chicago, IL, USA) using a one-way ANOVA and linear and quadratic analyses, and a Duncan’s multiple comparison test was used for descriptive analyses and difference significance tests. The replicates of 30 pairs of pigeons were the experimental units for all the data presented, and the differences were assumed to be statistically significant when *p* < 0.05.

## 3. Results

### 3.1. Productive Performance

Body weight gain, daily feed intake, laying rate, average egg weight, and feed to egg ratio are shown in Table 2. Body weight gain and daily feed intake for the whole experimental period were not affected by dietary Ca levels (*p* > 0.05). Laying rate, average egg weight, and the feed to egg ratio were significantly influenced by dietary Ca levels (*p* < 0.05), with all of them showing significant quadratic responses to increasing dietary Ca levels (*p* < 0.05) with a regression equation of y = −0.3945x^2^ + 0.8422x + 4.6256 (maximum response = 1.07), y = 0.6491x^2^ − 2.5501x + 24.313 (minimum response = 1.96) and y = 1.5364x^2^ − 4.226x + 12.665 (minimum response = 1.38). According to the laying rate and feed to egg ratio, the optimal dietary Ca level was 1.23%.

### 3.2. Egg Quality

The percentage of pigeon egg components and conventional egg quality indexes are shown in Table 3. Albumen percentage, albumen height, the Haugh unit and eggshell thickness at 8 w and 16 w were significantly influenced by dietary Ca levels (*p* < 0.05), all showing significant linear responses to decreasing dietary Ca levels (*p* < 0.05). All of them displayed downward trends when dietary Ca levels were decreased. Eggshell percentage, yolk percentage, egg shape index and yolk color at 16 w were affected by dietary Ca levels (*p* < 0.05). The egg shape index showed a significant linear response (*p* < 0.05), and the eggshell percentage, yolk percentage, egg shape index and yolk color all showed significant quadratic responses (*p* < 0.05) with a regression equation of y = 0.3086x2 − 1.2772x + 9.3781 (minimum response = 2.20), y = −1.3766x^2^ + 4.1111x + 20.423 (maximum response = 1.49) and y = 0.6033x2 − 2.3138x + 5.9922 (minimum response = 1.92). However, eggshell color and eggshell strength were not significantly influenced by dietary Ca levels (*p* > 0.05). When dietary Ca levels decreased, the eggshell and albumen quality also tended to decrease.

### 3.3. Albumen Transparency

Table 4 shows that the *L**, *a**, *b**, and *c** values of cooked albumen at 8 w and 16 w were all significantly influenced by dietary Ca levels (*p* < 0.05), and all showed significant linear responses to decreasing dietary Ca levels (*p* < 0.05). The *L** and *a** values showed a downwards trend when the dietary Ca level decreased, and the *b** and *c** values showed opposite trends. The distribution of transparent, translucent and opaque eggs between different Ca levels at 8 w and 16 w are shown in Figure 1 and Figure 2. During the middle portion of the experiment, any trends regarding the percentage of transparent, translucent and opaque eggs were not obvious by the decrease of dietary Ca levels, but after 16 w of feeding different Ca levels, the percentage of transparent eggs presented an early increasing and later decreasing trend, and the percentage of opaque eggs showed an opposite trend. According to the *L** values and the distribution of different egg transparencies, the optimal dietary Ca level for the best albumen transparency was 0.90%.

## 4. Discussion

Ca is an important nutritional indicator for laying fowl and plays a vital role in poultry health and performance. The NRC suggested a Ca requirement of 3.6% for Brown laying hens at a feed consumption rate of 110 g/bird per day [7]. Moreover, Castillo et al. reported that the optimum biological Ca levels for maximum egg production were 4.38% and 4.64% in the diets of laying hens [11]. These percentages give the best feed conversion and maximum specific gravity. However, the Ca demands of pigeons still refer to the demand of chickens in practice until now, because there is no standard for pigeon nutritional requirements. There are many differences between hens and pigeons, including physiological characteristics, reproductive characteristics, and egg production rules. The results of our study showed that a dietary level of 1.23% was best for laying pigeon production. This was much lower than the recommended levels for laying hens. When the dietary Ca level was raised to 3.00%, the laying rate and feed to egg ratio declined dramatically. This may have been caused by metabolic disorders [17]. Using the recommended dietary Ca contents for hens for laying pigeons is not advisable, as it can cause the waste of feed ingredients and metabolic illnesses, and can also reduce production performance. 

Increasing the dietary Ca level to 3% did not cause a significant difference in body weight gain. This result agrees with Abou et al. and Chowdhury and Smith who reported that Ca levels of up to 4% in the layer diet had no significant effect on the final body weight [18,19]. The daily feed intake for the whole experimental period was not affected by dietary Ca levels and was similar to results obtained by Kermanshahi and Habavi, who stated that dietary Ca levels did not have a significant effect on feed consumption [20]. The energy and protein contents of the diet for laying poultry are the most important nutritional factors for ingestion and growth, but the distinction of Ca content within certain limits will not make a large difference on them. Low dietary Ca levels for laying poultry affect egg production and are an issue that is recognized by researchers [21,22]. Some research has also stated that a high Ca diet of 5% has no effect on egg production [23], while other studies have shown that high dietary Ca inhibits egg production in laying hens [12,24]. Based on our study, we hypothesize that too high or too low levels of dietary Ca will both affect egg production and that the optimal dietary Ca level is 1.23%. Simultaneously, we found that the laying rate of pigeons was at a very low level, which is because of the low fertility that it spent more than ten days for pigeon pairs to lay one nest eggs. The average daily feed intake had no significant difference when the dietary Ca level was different, but the feed to egg ratio showed significant difference because of the significant change of laying rate.

Studies regarding the effects of dietary Ca on egg quality were mainly concerned with the quality of the eggshell. Ca is the main mineral component of eggshells and is also responsible for internal egg quality. Eggshell quality is a vital factor in poultry production because large numbers of eggs with defective shells can lead to great economic losses. Jiang et al. reports that animals fed low dietary Ca had weaker eggshells and poorer bone quality than hens fed a control diet [25]. Daniele et al. found that increasing dietary Ca levels in Japanese quails directly influenced the percentage of broken eggshells, and the shell thicknesses were increased with Ca treatment up to the 3.85% [26]. The present study’s conclusion is similar to those from previous studies that state that eggshell thickness in laying pigeons was linearly increased by raising dietary Ca. In this study, we also found that Ca had a great effect on the quality of the egg albumen. Albumen percentage, albumen height and the Haugh unit were all linearly increased by raising dietary Ca. These results agree with Wu et al. who reported that improving Ca (3.86%, 4.00%, and 4.18%) and other nutrients significantly increased albumen weight [27]. However, there are some inconsistent studies, which propose that dietary Ca levels have no effect on yolk or albumen [26,28].

The albumen of uncooked pigeon eggs is clear and transparent. The change in transparency only occurs when the albumen is cooked because it contains a high concentration of protein, and the albumen is the portion of the protein dispersed in water. When it is heated, the solution can be converted into gelatin. Protein gelation is a complex process, and the differences in albumen transparency are caused by a combination of internal and external factors. The internal factors mainly include water content, protein concentration [29], composition of amino acids [30], composition and content of metal ions [30], isoelectric point and pH. The external factors are mainly the temperature and time of heating [31]. To date, there have been many studies focused on the optical properties of protein gelatin, but few have been conducted with a focus on the transparency of pigeon egg albumen. Our preliminary research showed that there were no significant differences in water content and protein concentration between transparent eggs and opaque eggs. The mineral elements (Na, K, Ca, Mg, Fe, Cu, and Zn) of transparent eggs were lower than in opaque eggs, and the content of Ca showed a significant difference between the two [6]. The content of Ca is speculated to be one of the main factors affecting pigeon egg transparency.

Our results showed that the *L** and *a** values of the albumen showed downward trends when dietary Ca levels were decreased, and the *b** and *c** values showed increasing trends. This means that the colour of the albumen was darker, greener, and yellower, and seemed fresher with decreasing Ca levels. We verified that decreasing dietary Ca levels can improve the transparency of pigeon eggs. Egg cooking is a protein denaturation process whereby the monovalent ions (Na^+^, K^+^) cause the protein molecular aggregation structures to become relatively dense, mainly through electrostatic interactions. The divalent ions, such as Ca^2+^, also make three-dimensional net structures that are relatively dense by forming ion bridges with negatively charged protein molecules [32]. These ion bridges make the protein molecules more concentrated, causing the lower light transmission of gelatin and ultimately forming eggs with opaque albumen. Therefore, by decreasing dietary Ca levels, the Ca deposits in pigeon eggs are reduced and the ion bridges in the eggs form less when cooking, causing the transparency of the pigeon eggs to become improved. According to distributions of different transparencies of the eggs, the optimal dietary Ca level was 0.90%, and the production performance between 0.90% and 1.20% of Ca had no significant difference. This suggests that a dietary Ca level of 0.90% is an appropriate level for feed producers to formulate into a professional feed for laying pigeons with transparent eggs.

## 5. Conclusions

(1)Moderate dietary Ca contents can increase the laying rate and feed conversion ratio, improve eggshell thickness and albumen quality, and enhance the egg transparency of laying pigeons.(2)The level of dietary Ca is 1.20% which is best for production performance. However, using the transparency of pigeon eggs as an assessment index, the optimal level of dietary Ca is 0.90% for laying White King pigeons.

## Figures and Tables

**Figure 1 animals-09-00110-f001:**
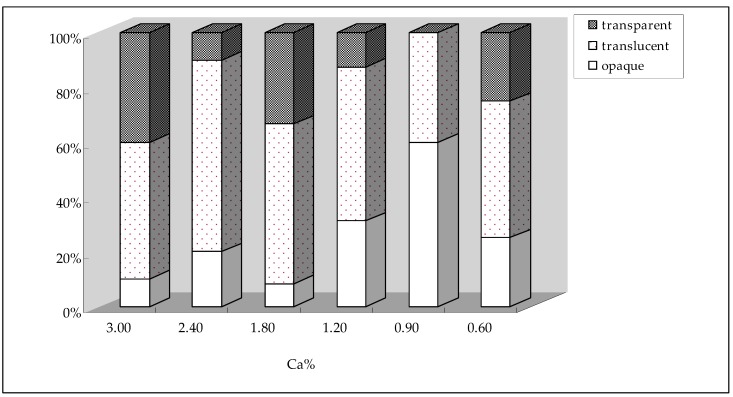
The distribution of transparent, translucent and opaque eggs between different Ca levels at 8 w.

**Figure 2 animals-09-00110-f002:**
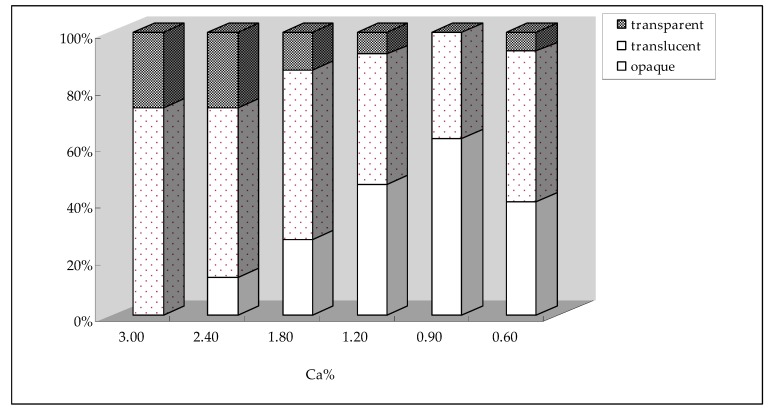
The distribution of transparent, translucent and opaque eggs between different Ca levels at 16 w.

**Table 1 animals-09-00110-t001:** Composition and nutrient levels of the diets.

Ingredient (%)	Group 1	Group 2	Group 3	Group 4	Group 5	Group 6
Corn	65.00	65.00	65.00	65.00	65.00	65.00
Soybean meal	15.57	15.57	15.57	15.57	15.57	15.57
Soybean oil	0.50	0.50	0.50	0.50	0.50	0.50
Compound Amino Acids ^1^	0.23	0.23	0.23	0.23	0.23	0.23
Premix ^2^	8.71	8.71	8.71	8.71	8.71	8.71
Ground rice hulls	0.00	2.08	4.18	6.29	7.31	8.37
Calcium hydrophosphate	4.05	3.11	2.16	1.20	0.74	0.26
Stone powder	5.94	4.80	3.65	2.50	1.94	1.36
Total	100.00	100.00	100.00	100.00	100.00	100.00
**Calculated Level ^3^**						
ME MJ/kg	11.15	11.15	11.15	11.15	11.15	11.15
CP %	13.00	13.00	13.00	13.00	13.00	13.00
Methionine %	0.21	0.21	0.21	0.21	0.21	0.21
Lysine %	0.75	0.75	0.75	0.75	0.75	0.75
Ca %	3.00	2.40	1.80	1.20	0.90	0.60
Total P %	0.98	0.82	0.66	0.50	0.42	0.34
Available P %	0.81	0.65	0.49	0.32	0.24	0.16
**Analyzed Level ^4^**						
Total E MJ/kg	10.98 ± 0.17	11.17 ± 0.27	11.07 ± 0.16	10.97 ± 0.20	11.04 ± 0.08	11.06 ± 0.20
CP %	12.89 ± 0.22	12.85 ± 0.24	12.79 ± 0.32	12.81 ± 0.40	12.82 ± 0.16	12.81 ± 0.23
Ca %	2.97 ± 0.13	2.36 ± 0.06	1.76 ± 0.12	1.14 ± 0.08	0.87 ± 0.12	0.58 ± 0.07
Total P %	0.93 ± 0.04	0.78 ± 0.05	0.62 ± 0.05	0.48 ± 0.03	0.40 ± 0.02	0.31 ± 0.02

^1^ Compound Amino Acids provided the following: methionine, lysine, tryptophan and threonine. ^2^ Premix provided the following: decavitamin, complex minerals, baking soda, choline chloride, salt, secondary powder, bentonite, et al. ^3^ Values were calculated from data provided by Feed Database in China (2010). ^4^ Values were presented as the means of triplicate per sample.

**Table 2 animals-09-00110-t002:** The effect of dietary Ca level on performance of laying White King pigeons.

Item	Ca Supplemental Level (%)						SEM	*p*-Value		
	3.00	2.40	1.80	1.20	0.90	0.60		ANOVA	Linear	Quadratic
Body weight gain (g)	−2.38	−14.67	−8.67	−14.17	−3.14	9.68	2.635	0.065	0.110	0.014
Daily feed intake (g/day)	36.85	36.74	35.65	37.29	36.00	35.74	2.347	0.833	0.520	0.878
Laying rate (%)	6.35 ^a^	6.57 ^a^	8.04 ^b^	9.76 ^c^	8.67 ^bc^	7.60 ^ab^	0.250	0.000	0.001	0.000
Average egg weight (g)	22.34 ^abc^	22.05 ^bc^	22.45 ^abc^	21.28 ^b^	22.47 ^ac^	23.42 ^a^	0.173	0.023	0.113	0.018
Feed to egg ratio (g/g)	13.71 ^a^	11.64 ^ab^	9.87 ^b^	9.73 ^b^	10.16 ^b^	10.71 ^b^	0.332	0.002	0.004	0.002

Different letters indicate significant differences (*p* < 0.05).

**Table 3 animals-09-00110-t003:** The effect of dietary Ca level on egg quality of laying White King pigeons.

Item			Ca Supplemental Level (%)						SEM	*p*-Value		
			3.00	2.40	1.80	1.20	0.90	0.60		ANOVA	Linear	Quadratic
8 w	Eggshell percentage (%)		7.42	7.46	7.56	7.61	8.11	8.09	0.698	0.091	0.004	0.591
	Albumen percentage (%)		71.70 ^a^	68.43 ^b^	69.51 ^ab^	68.01 ^b^	69.25 ^b^	68.46 ^b^	2.395	0.025	0.025	0.080
	Yolk percentage (%)		20.56	21.38	19.30	20.15	21.33	21.28	1.448	0.119	0.269	0.087
	Egg shape index		1.33	1.35	1.34	1.36	1.34	1.38	0.049	0.323	0.096	0.812
	Eggshell color	*L**	88.42	87.41	87.86	87.48	87.32	87.31	1.744	0.637	0.172	0.606
		*a**	−0.22	−0.43	−0.20	−0.44	−0.37	−0.59	0.352	0.071	0.026	0.420
		*b**	2.53	2.41	2.17	2.41	1.33	1.55	1.535	0.343	0.044	0.704
	Eggshell thickness (mm)		0.263 ^a^	0.228 ^b^	0.232 ^b^	0.226 ^b^	0.225 ^b^	0.226 ^b^	0.019	0.000	0.000	0.000
	Eggshell strength (kg/cm^2^)		1.105	1.111	1.126	1.137	1.100	1.066	0.142	0.934	0.563	0.379
	Yolk color (RYCF)		4.30	4.70	4.56	4.00	4.60	4.18	0.101	0.304	0.477	0.621
	Albumen height (mm)		3.07 ^a^	2.60 ^abc^	2.81 ^ac^	2.80 ^abc^	2.28 ^b^	2.45 ^bc^	0.634	0.055	0.012	0.862
	Haugh unit		71.99 ^a^	69.08 ^abc^	69.76 ^ac^	69.86 ^abc^	63.94 ^b^	64.69 ^bc^	6.893	0.038	0.003	0.656
16 w	Eggshell percentage (%)		8.14 ^bc^	8.51 ^ab^	7.92 ^c^	8.16 ^bc^	8.34 ^abc^	8.91 ^a^	0.655	0.028	0.047	0.040
	Albumen percentage (%)		72.07 ^ab^	71.16 ^bc^	73.14 ^a^	72.28 ^ab^	70.56 ^c^	70.63 ^c^	1.515	0.011	0.011	0.048
	Yolk percentage (%)		20.13 ^a^	23.07 ^b^	22.61 ^b^	24.01 ^b^	22.41 ^ab^	22.63 ^b^	2.383	0.043	0.078	0.018
	Egg shape index		1.23 ^a^	1.36 ^ab^	1.33 ^ab^	1.37 ^ab^	1.38 ^ab^	1.41 ^ab^	0.105	0.001	0.000	0.299
	Eggshell color	*L**	87.77	87.05	88.51	88.36	86.90	87.34	1.658	0.258	0.532	0.183
		*a**	−0.27	−0.37	−0.12	−0.30	−0.48	−0.53	0.329	0.266	0.042	0.212
		*b**	3.86	1.68	2.52	2.78	2.14	3.19	1.894	0.104	0.721	0.106
	Eggshell thickness (mm)		0.236 ^a^	0.224 ^b^	0.226 ^abc^	0.221 ^bc^	0.210 ^c^	0.214 ^bc^	0.016	0.002	0.000	0.556
	Eggshell strength (kg/cm^2^)		1.140	1.174	1.257	1.185	1.079	1.081	0.180	0.473	0.195	0.157
	Yolk color (RYCF)		4.40 ^b^	4.25 ^b^	3.33 ^c^	4.00 ^bc^	5.00 ^a^	4.50 ^ab^	0.099	0.001	0.034	0.006
	Albumen height (mm)		3.04 ^a^	2.70 ^abc^	2.88 ^ac^	2.57 ^abc^	2.16 ^b^	2.45 ^bc^	0.571	0.034	0.005	0.660
	Haugh unit		71.37 ^a^	68.88 ^ac^	69.77 ^ac^	67.55 ^abc^	63.26 ^b^	66.07 ^bc^	5.113	0.028	0.003	0.731

Different letters indicate significant differences (*p* < 0.05). RYCF is the abbreviation of regression between Roche yolk color fan.

**Table 4 animals-09-00110-t004:** The effect of dietary Ca level on albumen transparency of laying White King pigeons.

Item		Ca Supplemental Level (%)						SEM	*p*-Value		
		3.00	2.40	1.80	1.20	0.90	0.60		ANOVA	Linear	Quadratic
8 w	*L**	55.47 ^a^	52.00 ^ac^	50.97 ^ac^	46.77 ^bc^	43.62 ^b^	49.84 ^ac^	1.018	0.014	0.004	0.057
	*a**	−4.78 ^a^	−3.86 ^ac^	−4.64 ^a^	−3.76 ^ac^	−2.14 ^b^	−2.73 ^bc^	0.210	0.000	0.000	0.598
	*b**	−4.89 ^a^	−5.64 ^a^	-6.03 ^a^	−6.04 ^a^	−8.73 ^b^	−8.70 ^b^	0.298	0.000	0.000	0.327
	*c**	6.84 ^a^	7.27 ^a^	7.67 ^ab^	6.66 ^a^	9.23 ^b^	9.19 ^b^	0.270	0.007	0.002	0.251
16 w	*L**	52.54 ^a^	51.13 ^ab^	49.53 ^ab^	47.47 ^bc^	43.06 ^c^	47.67 ^bc^	1.837	0.009	0.000	0.196
	*a**	−4.60 ^b^	−4.58 ^b^	−3.11 ^ac^	−4.03 ^cb^	−2.47 ^a^	−2.82 ^ac^	0.222	0.013	0.005	0.246
	*b**	−6.21 ^a^	−8.51 ^ab^	−10.27 ^b^	−8.81 ^b^	−9.69 ^b^	−9.96 ^b^	0.370	0.011	0.004	0.076
	*c**	7.31 ^a^	9.46 ^b^	10.95 ^b^	9.33 ^ab^	11.04 ^b^	10.55 ^b^	0.337	0.007	0.004	0.081

Different letters indicate significant differences (*p* < 0.05).

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
