# Peer review of "Effects of Different Dietary Calcium Levels on the Performance, Egg Quality, and Albumen Transparency of Laying Pigeons"

_animals, 2019, doi:10.3390/ani9030110_

Round 1
Reviewer 1 Report
Review of the paper for the Animals journal
The aim of the research was to determine the effect of calcium level on productive performance, egg quality and albumen transparency. The manuscript presented for review brings new elements to the current state of knowledge defined in its subject. The research material is in sufficient numbers. Applied research methods correct. Before printing, the manuscript requires minor revision. The list of proposed changes is given below:
L19 1080 pigeons or 1080 pigeon pairs (see L69)
L20 30 pigeon pairs or 30 pigeons (15 pairs)?
L24 also eggshell percentage, yolk percentage, egg shape index, yolk colour at 16 w were significantly influence by calcium levels.
L28 In my country it is 1.0% Ca, 13% CP, 0.55% Liz, 0.26% Met, 0.6% Total P, 0.4% available P according to nutritional guidelines of 2018
L59 „Calcium suplement” or calcium level in the diet?
L71 "per replication" or "per treatment"?
L90 + No information about the scales (type, name, manufacturer's details) and measurement accuracy for the characteristics of BW, FI, eggshell, yolk, albumen weight
L97 Total 360 pigeon eggs evaluated?
Table 2 No significance marks at 22.34 and 22.45 for "average egg weight"
L235: eggshell thickness or eggshell strength?
L238: 0.9? or 1.2? for productive performance
Author Response
Dear Professor :
Thank you for your kind suggestions on this manuscript, and here below is our description on revision according to your comments.
Chang lingling
2019-3-9

Reviewer 2 Report
The study reported in paper deals with the topic and species that has received limited attention in the literature. The results are of interest and a good contribution to current knowledge on pigeon nutrition. However, this reviewer has some concerns and a major revision will be required.
The results have been correctly analysed using orthogonal polynomials, which is pleasing.
MAJOR COMMENTS
L78 Please tell why this ratio was used?. What was the rationale?.
Table 2: Check the BW gain unit – cannot be kg???; use g/d; BW gain only at 0.60% Ca, but no difference in feed intake. It tells something, but not discussed in the text. Laying rate of 6-8% - is this standard? – how did you calculate laying rate% - show the formula after L83.. Looks very low. Please comment in the discussion
L169 This statement is wrong – see Table 2. Also discuss this and provide a good explanation. L170- L182: discussion must be rephrased accordingly.
L181 but the birds are losing weight. Such a low Ca level contradicts everything that we know about Ca level in laying poultry diets. A detailed explanation is required. What is the current Ca levels used in such diets?. How do you explain your results?
MINOR COMMENTS
Abbreviate ‘calcium’ to ‘Ca’ in the textL55
L20 …0.60, 0.90 … and 3.00% were fed.
L42 Change ‘w’ to wks or weeks
L34 ..dish? rephrase
L47 .Some research exists …
L54 A deficiency of Ca
L55 ..cause skeletal abnormalities
Table 1: use % Ca level, instead of Groups 1 to 6; Rice hull powder – should this be Ground rice hulls?; Ca hydrophosphate or DCP?; Stone powder or fine sand?; How did you analyse for ME? – not possible.
L123 But the birds are losing weight at 1.23% Ca?
Table 3: for clarity, please separate 8 wk and 16 wk data within this Table. Same for Table 4.
Figures 1 and 2: X axis must be % Ca level, not 1,2 ….6.
Author Response

(The authors gave the same response as above.)
